# Designing Ecological Auditory Feedback on Lower Limb Kinematics for Hemiparetic Gait Training

**DOI:** 10.3390/s23083964

**Published:** 2023-04-13

**Authors:** Prithvi Ravi Kantan, Sofia Dahl, Helle Rovsing Jørgensen, Chetali Khadye, Erika G. Spaich

**Affiliations:** 1Department of Architecture, Design and Media Technology, Aalborg University, 2450 Copenhagen, Denmark; 2Neuroenhed Nord, Regionhospital Nordjylland, 9700 Brønderslev, Denmark; 3Division of Population Health and Genomics, University of Dundee, Dundee DD1 4HN, Scotland, UK; 4Department of Health Science and Technology, Aalborg University, 9260 Gistrup, Denmark

**Keywords:** biofeedback, swing phase, acoustic feedback, gait rehabilitation, assistive technology, hemiparetic, gait

## Abstract

Auditory feedback has earlier been explored as a tool to enhance patient awareness of gait kinematics during rehabilitation. In this study, we devised and tested a novel set of concurrent feedback paradigms on swing phase kinematics in hemiparetic gait training. We adopted a user-centered design approach, where kinematic data recorded from 15 hemiparetic patients was used to design three feedback algorithms (wading sounds, abstract, musical) based on filtered gyroscopic data from four inexpensive wireless inertial units. The algorithms were tested (hands-on) by a focus group of five physiotherapists. They recommended that the abstract and musical algorithms be discarded due to sound quality and informational ambiguity. After modifying the wading algorithm (as per their feedback), we conducted a feasibility test involving nine hemiparetic patients and seven physiotherapists, where variants of the algorithm were applied to a conventional overground training session. Most patients found the feedback meaningful, enjoyable to use, natural-sounding, and tolerable for the typical training duration. Three patients exhibited immediate improvements in gait quality when the feedback was applied. However, minor gait asymmetries were found to be difficult to perceive in the feedback, and there was variability in receptiveness and motor change among the patients. We believe that our findings can advance current research in inertial sensor-based auditory feedback for motor learning enhancement during neurorehabilitation.

## 1. Introduction

Persons who have sustained a cerebral stroke or traumatic brain injury commonly exhibit predominantly one-sided weakness (hemiparesis), and the patterns (spatiotemporal, kinematic, and electromyographic) that characterize their walking are broadly termed as hemiparetic gait [1]. Gait rehabilitation aims to reinstate gait function to its former level by repetitive training, a process that leverages neuroplasticity and essentially involves ‘re-learning’ [2] how to walk as before. In a very broad sense, *feedback* is crucial to movement execution as well as learning processes. In normal motor execution, the central nervous system integrates information from multiple feedback channels (e.g., vision, proprioception, and touch) into the motor control processes underpinning gait [3], and proprioception is especially important in motor planning and coordination [4]. Feedback is also an indispensable component of learning in general [5] and motor learning in particular [6], and its effectiveness depends on several factors. In classroom learning, it has been established that students can only respond to timely and relevant feedback that they can understand and are capable of acting on, and which is ideally constructive and tempered with a sense of positivity [5]. When it comes to designing biofeedback applications for motor rehabilitation, the interplay between several factors (mode, content, frequency, timing) determines overall effectiveness, and some of these mechanisms are not well understood [6,7,8].

In recent years, the auditory modality has been increasingly explored as a route to provide biomechanical biofeedback during gait rehabilitation [9,10,11]. As this typically works by conveying movement data as auditory perceptual variations, it is a case of *interactive sonification of movement* [11,12]. Several factors make the auditory modality suitable for providing real-time feedback on walking. Hearing has fine temporal resolution [13] and can free a user from additional visual burden and distraction [6]. In addition, research has shown strong connections between the auditory and motor areas of the brain [14] as well as evidence of common neural coding of auditory and motor patterns [15]. Moreover, it has been shown that based on sound alone, humans have a natural ability to perceive and reenact the spatiotemporal characteristics of complex movements such as walking [16]. Thus, the use of sonification helps create an enhanced multimodal perceptuomotor workspace for practicing movements [17]. Through multisensory integration, a set of semantically congruent and synchronous stimuli representing the movement can evoke a stronger neural response than the most effective individual stimulus [18,19]. Multisensory learning has been shown to confer learning benefits that are retained even during subsequent unisensory task performances [20].

Past studies on gait feedback have typically involved systems that informed users on spatiotemporal parameters (e.g., cadence, step length), stance phase-specific variables (e.g., ground reaction forces), and joint angles (e.g., knee flexion and extension during the stance or swing phase) [10,21,22,23]. A relatively small proportion of existing investigations have provided explicit feedback on the *swing phase* (non-ground contact). A good example is by Rodger et al. [24], who sonified instantaneous limb swing velocity in Parkinson’s disease patients as musical pitch variations, finding that it led to, as hypothesized, a significant reduction in step length variability. Giraldo-Pedroza et al. [25] found that threshold-based haptic feedback on swing time was able to induce significant changes in stride length, swing time, and hip flexion in healthy older adults (who typically take short steps with a reduced hip motion range). Hemiparetic patients can exhibit considerable swing phase asymmetry due to reduced joint motion ranges and compensatory strategies (e.g., circumduction) prevalent on the most-affected side [1]. The vast majority of stroke patients suffer from tactile and proprioceptive somatosensory deficits, which are known to influence variance in stride length, gait velocity, and walking endurance [26]. It is well-documented that proprioceptive information during the swing phase is of considerable functional importance during locomotion [27], and that improved somatosensory feedback can generally lead to more accurate timing and amplitude of muscle contractions in response to the external environment [28]. Concurrent feedback on leg kinematics during the swing phase may therefore have assistive potential in gait training, but has not, to the best of our knowledge, been widely explored with this patient group.

The clinical potential of gait biofeedback is yet to be concretely established due to inconsistencies in the methods used in existing research [8,9,29,30,31]. Key issues include the lack of proper understanding of the underlying neurophysiological, biomechanical, and motor learning mechanisms of gait biofeedback [8] and the lack of integration of motor learning principles into biofeedback design [10,30]. Systematic reviews have suggested that studies should describe their feedback rationale in detail [7] and focus on direct comparisons between different modes of feedback [31]. An important consideration specific to the auditory modality is how an otherwise-silent movement should be represented through sound so as to maximize semantic congruence between the interacting modalities. One solution is to adopt meaningful sound structures that are directly or metaphorically associated with the movement in question. Such sonification approaches have been recommended for scientific tasks in general by Neuhoff [32], who proposed the use of sounds that are "acoustically complex but ecologically simple”. Specific to motor learning, it has been suggested that people already know how to engage with certain sound morphologies in certain contexts (e.g., splashing around in water), and that these morphologies can make for more intuitive feedback systems than the typically used abstract morphologies (e.g., synthesized sine waves) [17]. Moreover, there is evidence that ecological sounds (e.g., naturalistic footstep sounds for feedback on the stance phase) afford a greater degree of feedback comprehension and movement modification than abstract sounds due to superior matching between these sounds and the previous experiences of the learner [33,34,35]. The aim of this exploratory study was to devise and test inertial sensor-based concurrent swing phase auditory feedback in the context of hemiparetic gait training.

## 2. Materials and Methods

### 2.1. User-Centered Design Methodology

Hemiparetic patients often exhibit considerable intra-group variability [1], which poses significant challenges to technology designs. Our study methodology was, therefore, rooted in the principles of user-centered design. It has long been recommended that patients and professionals be systematically and meaningfully involved in the development process of complex interventions and technologies so that these are customized to their needs [36,37]. The user-centered design incorporates user-centered activities throughout several iterations of development and testing [36,38,39] with the ultimate goal of increasing usability and user acceptance [40,41]. User-centered methods have been recommended for building sound-based health technologies as well [38]. Van Dijk’s methodological recommendations include investigating simple interventions and conducting small-scale qualitative evaluations to quickly and efficiently assess basic ideas [42]. We integrated these principles into the development and feasibility evaluation of multiple *feedback algorithms (FAs)*. We started by recording motion sensor data from fifteen hemiparetic patients to serve as the basis for developing three FAs, which were evaluated first-hand by a focus group of five experienced physiotherapists. Based on their recommendations, we discarded two algorithms and made some key modifications to the remainder. We then carried out a feasibility study with nine hemiparetic patients and seven physiotherapists, where our auditory feedback was applied as an add-on to conventional overground training paradigms, and the applicability of several feedback variants was assessed in terms of kinematic characterizations and subjective experience.

### 2.2. Feedback Algorithm Design Software

The practical development of FAs was centered around a custom-built sonification software program with the following main capabilities:1.Real-time inertial data reception and recordings from up to five inertial sensors—M5Stack Gray (manufactured by M5Stack, sourced from Copenhagen, Denmark) devices equipped with MPU 9250 9-axis inertial units (https://shop.m5stack.com/products/grey-development-core, accessed on 28 February 2023). The MPU-9250 has, in comparison with optical systems, been shown to exhibit sufficient validity and reliability for gait analysis purposes in healthy [43,44] and patient populations [45,46]. These sensors transmitted data ($f_{s}$ = 100 Hz) to the software over WiFi using the UDP protocol.2.Real-time sonification of movement features computed from the raw inertial data.3.Dedicated interface for real-time manipulation of the sonification algorithm (depicted in Figure 1).4.Storage and recall of sonification algorithms.5.Gait visualization from recorded and real-time data using stick figure representations.6.Sonification of the recorded inertial data at a user-specified rate to simulate real-time motion input.7.A range of inbuilt audio synthesizers implemented in FAUST (https://faust.grame.fr/, accessed on 28 February 2023) as well as the ability to send mapping data to third-party audio software using UDP.

The overall system was designed as a distributed biofeedback structure to maximize the available computational power for stimulus computation [47], Chapter 5. The software was built using the JUCE (https://juce.com/, accessed on 28 February 2023) environment for C++. The algorithm design (mapping) interface comprises a matrix layout with movement feature source selectors arranged along the row dimension and audio control signals (corresponding to synthesizer parameters) along the column dimension. The internal architecture treats all features as signals, normalizing their value ranges of interest to a common 0–1 range so they can be added and subtracted meaningfully. The matrix is made up of binary toggle buttons that enable or disable connections between the corresponding pairs of movement features and audio control signals, as well as sliders to adjust the strength of each connection. To pre-process the movement features prior to final mapping, there are several digital signal processing operations such as filtering, nonlinear transformation, step quantization, polarity inversion, and gain adjustment. The user can adjust all settings in real-time, including the movement feature range of interest and the audio signal feature range to suit individual patients’ needs. Hence, a sonification algorithm is realized as a unique configuration of the interface, which can be stored as a preset for future recall.

### 2.3. Feedback Rationale

In simple terms, the proprioceptive inputs from joint and muscle spindle receptors are integrated by the central nervous system to yield a unified sensation of body positioning and its temporal variations [4]. This is used by the central nervous system to adjust feedforward motor commands during gait, which in turn control muscular activity [3,4], resulting in rotations of the thigh, shank, and foot segments about the hip, knee, and ankle joints, respectively. Additionally, it is used to coordinate the bilateral movement of the legs, the lateral transfer of body weight, and the posture of the upper body [4]. The temporal characteristics of these segment rotations are, thus, tightly linked to the internal representation of the movement derived from proprioceptive input processing and the integration of other sensory inputs that inform, for instance, on the terrain and surroundings. As end-effector kinematics are believed to play a key role in motor control [48], we argue that by providing extrinsic feedback on segment rotations concurrently with the movement, the internal motor representation formed from proprioceptive inputs can be strengthened by multisensory integration [19,49], which can positively mediate feedforward control in subsequent steps and potentially result in sustained motor performance benefits [20]. As the characteristics of real-life movement-generated sounds (e.g., writing, musical instrument performance) are largely considered to be governed by the velocity profile of the excitatory movement [50], we decided to directly sonify the thigh and shank angular velocity profiles during the swing phase.

Thus, the central design challenge is to convert bilateral thigh and shank angular velocity signals into a feedback stimulus that is perceived as temporally, spatially, and semantically congruent with the natural proprioceptive input [18,49]. Ideally, there should not be an ’intellectual barrier’ or extra cognitive step involved in decoding movement characteristics from the stimulus. In other words, the user should hear the movement directly and implicitly understand its nature. In order to achieve this, there needs to be an intuitive and transparent metaphor determining the translation of kinematic information to sound [6,51,52]. Given that the constructs of speed and velocity are directly relatable to the energetic qualities of movement, we deemed that it would make sense to convey these constructs in the form of variations in the perceived energetic qualities of sound. Acoustically speaking, the audio signal properties correlated to movement energy are amplitude, spectral bandwidth, and tonal brightness. This metaphor lends itself well to the biomechanics of walking. As the swing phases alternate and do not overlap, the user is only informed of the movement of one limb at a time (the swinging limb) in an analogous alternating fashion. This can allow the user to synchronously and unambiguously link the motion of that limb to the sound accompanying its motion.

### 2.4. Feedback Requirements

Based on relevant conclusions from the literature, we framed a set of requirements that our auditory feedback technology needed to satisfy.

1.*Swing phase focus:* It must concurrently convey segment angular velocity information during the forward rotation of the thigh and shank in the swing phase.2.*Perceptual simultaneity:* The feedback must be perceived by the user to occur in synchronization with the movement with no noticeable latency [10,47].3.*Wireless wearable sensing and feedback actuation:* All motion sensing must be carried out using wireless wearable *inertial measurement units (IMUs)*, and the feedback should be delivered wirelessly to the patient using a suitable device (headphones or loudspeakers, depending on the training paradigm) [10].4.*Clinically compatible:* The technology and its application must be compatible with existing gait training protocols for hemiparetic patients, and necessitate little or no restructuring of current clinical processes. It must be versatile in order to suit the inherent diversity of motor and cognitive impairment in patients [1,8,10,38], in addition to being inexpensive and portable.5.*Feedback design:* The feedback must satisfy the following requirements for the target group:It must be capable of supporting motor learning without introducing cognitive barriers related to feedback interpretation:−There must be a direct mapping of movement dynamics to the sound [17,53,54].−The sound design must be ecological and action-consistent in such a way that the feedback directly elicits the perception of the movement [17,19,33,34,55].−The feedback must match the movement trajectory in time, space, and semantic content so as to support multisensory integration with proprioceptive sensations [18,49,56].It must be capable of enhancing patient motivation during training [5,9].It must be simple, intuitive, and meaningful [9,51,52].The sonic aesthetics must be acceptable to patients, and the feedback must be tolerable (not fatiguing) for the duration of a training session [9,10,52].

### 2.5. Motion Data Collection and Analysis

For kinematic analysis and feedback design, we obtained a gait dataset from 15 hemiparetic patients (9M, 6F, 65.53 ± 16.48 years old) during overground walking. All of them had unilateral weakness of varied severity due to ischemic or hemorrhagic strokes, intracranial hemorrhages, or traumatic brain injury. The dataset consisted of video footage as well as IMU recordings (100 Hz sampling rate) taken from the trunks, thighs, and shanks of patients walking in a straight corridor with their typical aids (physiotherapist support, rollators, training benches, etc.). Informed consent was obtained in advance and all procedures were conducted in accordance with the Declaration of Helsinki. The details of the data collection process are provided in [57]. We then conducted a preliminary graphical analysis of the IMU recordings in MATLAB R2022b. From the raw gyroscope readings corresponding to the sagittal plane, the thigh (T), and shank (S) *segment angular velocities (SAVs)* were computed as follows. First, the pre-calibrated static IMU bias was subtracted from the raw data values. To attenuate unwanted high-frequency components, the raw readings from the left (L) and right (R) thigh and shank were smoothed using second-order Butterworth low-pass filters (IIR, $f_{c}$ = 5 Hz, Q = 0.7). The cutoff frequency was empirically tuned so as to achieve the best trade-off between high-frequency rejection and phase delay. The polarity of the right side sensors was inverted so as to achieve a common sign convention (positive values = forward rotation) and the signal values were bounded between −300${}^{\circ }$ and 300${}^{\circ }$. The final values are, henceforth, referred to as $\omega_{TL}$, $\omega_{TR}$, $\omega_{SL}$, and $\omega_{SR}$.

We compared the observable trends in the plots with an analysis of the video footage by a physiotherapist. In general, we found that the patients’ gait impairments manifested as uniquely-shaped SAV trajectories as exemplified in Figure 2. The general gait asymmetry is seen as clear differences between the most- and least-affected sides for both P1 and P2 in the thigh and shank trajectories. For P1, compensation for the reduced hip *range of motion (RoM)* is shown by rapidly swinging modulations in the positive portion of the most-affected thigh velocity trace. A lack of control over the knee extension on the least-affected side is seen as the steep trailing edge of the shank SAV positive peaks. In the case of P2, more severe observed hip and knee kinematic issues on the most-affected side similarly manifested in the SAV trajectories. This early evidence of SAVs capturing hemiparetic gait patterns reinforced the case for using them directly as a concurrent feedback variable. We recreated the SAV computation in our real-time feedback software (along with a bias compensation feature) and computed two additional quantities:*Knee angular velocities:* The angular velocities of knee rotation were calculated as the instantaneous difference between the smoothed thigh and shank SAVs on either side. Signal polarity was adjusted to achieve a common sign convention (positive values = knee extension); we refer to these angular velocities as $\omega_{KL},\omega_{KR}$.*Step modulo:* This was a cyclically varying measure that kept track of the number of steps completed by the user. Steps were simply tracked from the local minima and maxima of the thigh SAV exceeding an adjustable threshold. This was calculated as:
(1)$$M_{step}=mod\left(\frac{N_{steps}}{S},D\right)\cdot x_{rand}\cdot \left(1+a_{rand}\right)$$
where the divisor *D* represents the number of discrete levels that $M_{step}$ could take (user-adjustable: 2, 4, 6, 8, 10, 12) and *S* represents the number of steps for which $M_{step}$ would remain at one value (user-adjustable: 1, 2, 4). $x_{rand}$ is a random number multiplier that could be applied to the modulo product (−1 to +1 range, refreshed after each completed step), controlled by $a_{rand}$, the amount/strength of introduced randomness (user-adjustable, 0–1 range). $M_{step}$ makes it possible to periodically trigger changes in the feedback characteristics after a given number of steps.

### 2.6. Developed Feedback Algorithms

We imported the patient data into the sonification software and developed an initial set of three FAs that transformed thigh, shank, and knee ω signals into a concurrent auditory feedback stream. The algorithms had the following traits in common:The ω values were normalized between 0 and 1, such that the rest state and backward segment rotations (negative initial values) corresponded to 0, the configurable patient-specific max ω for that segment corresponded to 1 and, thus, the forward rotations were in the (0, 1) range. These normalized signals ($\omega_{nTL}$, $\omega_{nTR}$, etc.) then underwent a series of preparatory operations such as smoothing, summing, nonlinear transformation, and linear gain to yield a parallel set of control signals $C_{1}$–$C_{3}$, which were bounded between 0 and 1 and then scaled to an appropriate value range for audio synthesizers and processors. All operations were defined using an analysis-by-synthesis approach. These operations took place at the sensor sampling rate of 100 Hz, and are shown on the left side of Figure 3, Figure 4 and Figure 5.The real-time audio output had a sampling rate of 48 kHz, a resolution of 24 bits, and a channel count of 2 (stereo).At any instant, immobile or backward-rotating limb segments were silent, whilst forward-rotating segments generated sounds governed by their corresponding ω values. It was expected that this mapping would keep the instantaneous focus of the user on the segment(s) that were in forward rotation. This could help avoid cluttering the auditory channel with non-informative sounds related to the stance phase limb. Moreover, L-R differences in thigh and shank SAV trajectories that were inherent to asymmetric gait patterns were readily reflected in the discrepant energetic properties of the resulting sounds.

Video demos of all feedback algorithms and their variations are provided in our online video repository (https://drive.google.com/drive/u/0/folders/1vgPPiKEblYRYPToTxLw4OdAG9S4_wRT8, accessed on 28 February 2023), and the software preset files are provided in the Appendix A.

#### 2.6.1. Feedback Algorithm 1 (FA1)—Wading

*Rationale:* The swing phase of gait, although normally silent in overground walking, generates sound when one wades through a shallow liquid. The properties of the generated sound are governed by the nature of the physical excitation (limb swing) and can, hence, be considered a real-life sonification of the swing phase that most people are familiar with from experience. Our feedback strategy aimed to leverage this experience by digitally creating a plausible real-time simulation of wading sounds. In principle, this was done by modeling their spectral composition after steady-state flowing water sounds (recorded from real-life sources), whose amplitude envelopes were controlled by the SAVs. Hence, liquid sounds were concurrently generated with limb movements (as in real wading). To alleviate monotony, we also made it possible to add a layer of ambient music to the wading sound. Optionally, the tonal characteristics of the music could be interactively controlled by knee rotation by mapping $\omega_{KL}$ and $\omega_{KR}$ to the center frequency of a peaking filter (bell-shaped frequency band emphasis) applied to it.

**Figure 3 sensors-23-03964-f003:**
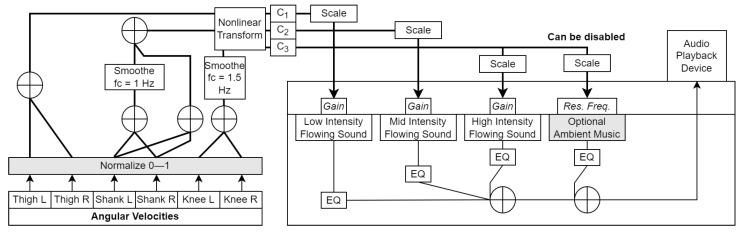
Block diagram of the wading algorithm. Grey boxes indicate modifiable parameters. *Res. Freq* refers to the center frequency of the peaking filter applied to the ambient music track. *EQ* refers to the spectral equalization applied to each sound for tone adjustment. $C_{1}$–$C_{3}$ are the audio parameter control signals generated from the movement features.

*Audio synthesis model:* The core model was made up of three recordings of steady-state flowing water downloaded from the *FreeSound* library (https://www.freesound.org, accessed on 28 February 2023). These were placed on parallel audio channels/tracks in REAPER (https://www.reaper.fm, accessed on 28 February 2023), a digital audio workstation software, and played back in an infinite loop. In terms of perceived energetic qualities, these ranged from *low intensity* (trickling) to *high intensity* (river flowing). Each of these underwent tonality adjustment through equalization and had a real-time modifiable gain parameter (-inf dB–0 dB, default -inf dB, or silence). Aside from the water sounds, it was optionally possible to add an ambient music track downloaded from Pixabay Music (https://pixabay.com/music/, accessed on 28 February 2023), which we equalized for minimum spectral overlap with the water sounds. This music was also processed by a peaking filter (gain = 12 dB, Q = 1.8) with modifiable center frequency for optional real-time tonality manipulation by the user. The signals were summed to a stereo output. All audio processing was done in REAPER.

*Control signal computation:*$C_{1}$ represented $\omega_{TL}$ and $\omega_{TR}$ (thighs), and was mapped to a low-intensity flowing sound (resembling trickling water). $C_{1}$ was computed as follows:
(2)$$\omega_{nT}=\omega_{nTL}+\omega_{nTR}$$
(3)$$C_{1}={\left(\omega_{nT}\right)}^{1.4}$$

$C_{2}$ represented $\omega_{SL}$ and $\omega_{SR}$ (shanks) and was mapped to a medium-intensity flowing sound. Similar to before, the normalized values $\omega_{nSL}$ and $\omega_{nSR}$ were summed to yield $\omega_{nS}$, which was then added to a smoothed version of itself ${\hat{\omega }}_{nS}$ (2nd-order Butterworth low-pass filter, $f_{c}$ = 1 Hz, Q = 0.7) to yield $C_{2}$ as:
(4)$$\omega_{nS}=\omega_{nSL}+\omega_{nSR}$$
(5)$$C_{2}={\left({\hat{\omega }}_{nS}+\omega_{nS}\right)}^{0.59}$$

The smoothing and summing were done in order to shape $C_{2}$, such that it would rise in immediate response to the shank rotation ($\omega_{nS}$ component), but would decay gradually (${\hat{\omega }}_{nS}$ component), resulting in the water sound slowly fading out rather than abruptly cutting out when the shank ceased to rotate forward (unnatural).

$C_{3}$ represented $\omega_{KL}$ and $\omega_{KR}$ (knees) and was mapped to a high-intensity flowing sound as well as (optionally) the center frequency of the peaking filter applied to an ambient music track. $\omega_{nKL}$ and $\omega_{nKR}$ were summed and smoothed (2nd-order Butterworth low-pass filter, $f_{c}$ = 1.5 Hz, Q = 0.7) to yield ${\hat{\omega }}_{nK}$, which was nonlinearly transformed to yield $C_{3}$ as:(6)$$\omega_{nK}=\omega_{nKL}+\omega_{nKR}$$
(7)$$C_{3}={\left({\hat{\omega }}_{nK}\right)}^{0.48}$$

$C_{1}$, $C_{2}$, and $C_{3}$ were all linearly scaled to a decibel range from −144 dB (silence) to +12 dB and sent to REAPER to control the synthesis model. If $C_{3}$ was also mapped to the filter center frequency, it was scaled to a range from 200 to 4000 (corresponding to Hz).

#### 2.6.2. Feedback Algorithm 2 (FA2)—Abstract Waveform

*Rationale:* This algorithm aimed to convey thigh and shank SAVs in a perceptually salient manner by manipulating the loudness, bandwidth, and resonant characteristics of a simple computer-generated periodic waveform with musical frequencies and a harmonically rich spectrum. The principle was that the intensity and bandwidth of the waveform would be controlled by a combination of thigh and shank rotation velocity, whereas its resonant qualities would only be controlled by the shank. We found that this configuration led to clearly distinguishable sounds when sonifying gait data with distinct SAV trajectory shapes. To give the patient the sense of driving a melody forward with their gait, the step modulo variable controlled fundamental frequency changes (notes of the A major scale), such that a new note played every two steps.

*Audio synthesis model:* The basic signal was a sawtooth waveform with modifiable fundamental frequency $f_{0}$ and a peak-to-peak amplitude of −1 to 1. The sawtooth was chosen for its periodic nature and harmonic richness. As shown in Figure 4, the waveform was passed through two cascaded IIR filters, i.e., a peaking/bell filter (Q = 3, +24 dB gain at a modifiable frequency), and a resonant low-pass filter (Q = 2, modifiable cutoff frequency). The center frequency of the peaking filter could be set between the fundamental frequency of the sawtooth and the cutoff frequency of the low-pass filter. The filter output was passed through a hard clipper (±0.1) to increase upper harmonic richness, followed by a modifiable overall gain multiplier. In all, the model exposed three parameters for real-time control, i.e., the *fundamental sawtooth frequency*, which altered the pitch of the sound; the *relative resonant frequency*, which altered the timbre of the sound in a vowel-like fashion by altering the peaking filter frequency relative to the fundamental, and the *dynamics coefficient*, which altered both the low-pass cutoff frequency (4.5$f_{0}$–14.5$f_{0}$, empirically determined) and the output gain multiplier (−inf dB–0 dB). To enhance pleasantness, the output was passed through a digital reverberator algorithm (a combination of the Schroeder and feedback delay network design [58], Chapter 14) to simulate a real space. All audio processing was conducted within the software using FAUST library functions.

**Figure 4 sensors-23-03964-f004:**
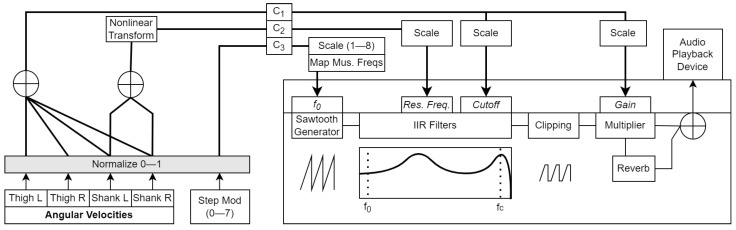
Block diagram of the abstract waveform algorithm. Grey boxes indicate modifiable parameters. *Res. Freq.* refers to the center frequency of the peaking filter applied to the sawtooth. $C_{1}$–$C_{3}$ are the audio parameter control signals generated from the movement features.

*Control signal computation:*$C_{1}$ represented $\omega_{TL}$, $\omega_{TR}$, $\omega_{SL}$, and $\omega_{SR}$ (thighs and shanks), and was mapped to the dynamics coefficient of the wave. It was simply computed as:(8)$$C_{1}=\omega_{nTL}+\omega_{nTR}+\omega_{nSL}+\omega_{nSR}$$

$C_{2}$ represented $\omega_{SL}$ and $\omega_{SR}$ (shanks only) and was mapped to the relative resonant frequency parameter. $C_{2}$ was calculated as:(9)$$\omega_{nS}=\omega_{nSL}+\omega_{nSR}$$
(10)$$C_{2}={\left(\omega_{nS}\right)}^{0.72}$$

Hence, the thighs and shanks both controlled the volume dynamics and bandwidth of the wave, but only the shanks controlled its tonal properties. $C_{3}$, on the other hand, was the normalized value of the step modulo $M_{step}$. $C_{1}$ was scaled from 0 to 0.7 prior to gain mapping, and from 4.5$f_{0}$ to 11.3$f_{0}$ prior to cutoff frequency mapping. $C_{2}$ was scaled from 0 to 0.52 prior to mapping; $C_{3}$ was scaled from 1 to 8 (representing note indices of a musical scale) and converted to musical frequencies in Hz (corresponding to the notes of the A major musical scale) prior to mapping.

#### 2.6.3. Feedback Algorithm 3 (FA3)—Synthesized Music

*Rationale:* The use of musical structures and spectra is known to have a positive effect on motivation and engagement during rehabilitation [59,60]. The goal of this strategy was to intuitively convey thigh and shank angular velocity using recognizable and distinct instrument textures in a musical pitch structure. As SAVs are continuously valued by nature, we chose to use instruments that, in real life, also received continuous physical excitation signals when played (the flute and violin). In short, the SAVs were linked to the excitation parameters of physics-based instrument simulations, and the movement information was conveyed concurrently through variations in the sound dynamics. To allow the patient to ‘drive’ the music forward with their gait, the fundamental frequencies of the instruments were similarly controlled by the step modulo variable, such that a new note was triggered every two steps. The note progression could either be fixed (ascending scale, such as FA2) or random. The flute and violin were separated by exactly one octave to minimize perceptual masking between them. It was also possible to add an optional non-interactive background music track to the sound generated by the patient.

*Audio synthesis model:* This was made up of physics-based simulations of flute and violin sounds, realized using a technique called waveguide synthesis [61]. In all, there were four real-time modifiable parameters: *flute fundamental frequency, flute blowing pressure, violin fundamental frequency,* and *violin bowing velocity*. Their summed output was passed through the same digital reverberation algorithm as in FA2. All audio processing was done within the software using FAUST. A background music track could be externally added.

**Figure 5 sensors-23-03964-f005:**
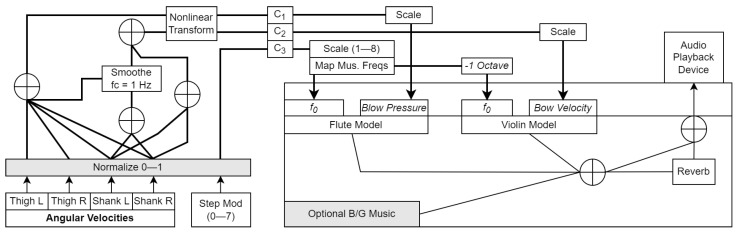
Block diagram of the synthesized music algorithm. Grey boxes indicate modifiable parameters. $C_{1}$–$C_{3}$ are the audio parameter control signals generated from the movement features.

*Control signal computation*: $C_{1}$ represented $\omega_{TL}$, $\omega_{TR}$, $\omega_{SL}$, and $\omega_{SR}$ (thighs and shanks), and was mapped to the blowing pressure of the flute. It was simply computed as:(11)$$\omega_{sum}=\omega_{nTL}+\omega_{nTR}+\omega_{nSL}+\omega_{nSR}+{\hat{\omega }}_{nS}$$
(12)$$C_{1}={\left(\omega_{sum}\right)}^{0.49}$$where ${\hat{\omega }}_{nS}$ was a smoothed version of $\omega_{nS}$ (fc = 1 Hz). $C_{2}$ represented $\omega_{SL}$ and $\omega_{SR}$ (shanks only) and was mapped to the bowing velocity of the violin. $C_{2}$ was calculated as(13)$$\omega_{nS}=\omega_{nSL}+\omega_{nSR}$$
(14)$$C_{2}={\left(\omega_{nS}+{\hat{\omega }}_{nS}\right)}^{1.87}$$

Hence, the thighs and shanks both controlled the dynamics of the flute, and only the shanks controlled the violin. Similar to the previous algorithm, $C_{3}$ was the normalized value of the step modulo $M_{step}$, which could be configured to increment monotonically or change in a more random manner. It was scaled as before to a 1–8 range and mapped to musical frequencies within the A major scale. These values were mapped to the flute and violin frequency parameters as shown in Figure 5.

## 3. Hands-on Testing by Physiotherapists

In order to obtain an expert assessment of the clinical potential of the FAs, we conducted a hands-on testing session with a focus group of physiotherapists, where we demonstrated the FAs to them, allowed them to try out each FA in real life, and conducted structured group interviews. The primary purpose of the session was to help us identify and hone the FA(s) with the most potential to be understood and tolerated by hemiparetic patients and, if necessary, scrap those that had serious design issues, which could not be addressed by simple modifications.

### 3.1. Participants

Five physiotherapists (1M, 4F—all specializing in neurorehabilitation) at Neuroenhed Nord, Regionhospital Nordyjlland, Denmark, with 14.4 ± 12.6 years of clinical experience, volunteered to participate in the interview. Informed consent was obtained in advance and all procedures were conducted in accordance with the Declaration of Helsinki.

### 3.2. Setup

The testing was carried out in a large meeting room at Neuroenhed Nord, Regionhospital Nordjylland, Brønderslev, Denmark. The biofeedback system was set up to run in real-time on a Lenovo P14s (manufactured by Lenovo, sourced from Copenhagen, Denmark) laptop through an M-Audio M-Track Solo audio interface (manufactured by M-Audio, sourced from Mumbai, India) and IKEA Eneby loudspeaker (manufactured by IKEA, sourced from Copenhagen, Denmark) (https://www.ikea.com/us/en/p/eneby-bluetooth-speaker-black-gen-2-10492403/, accessed on 28 February 2023). A TP-Link Archer C20 wireless router (manufactured by TP Link, sourced from Copenhagen, Denmark) was used to host a WiFi network over which the sensors transmitted data to the software. A Lenovo Yoga tablet (manufactured by Lenovo, sourced from Amazon India) was used to collect audio recordings of the interview. The room was arranged such that there was a sitting area for briefing, explanations, and discussion, as well as an open floor area for walking and a real-life tryout of the FAs.

### 3.3. Procedure

The duration of the session was two hours in all. It began with a 15-min introduction of the project and general feedback philosophy. Following this, each feedback algorithm was demonstrated and discussed in a similar format. We spent approximately 30 min per algorithm (order FA2–FA1–FA3), starting with a 5-min explanation of each concept with video demos showing how the feedback sounded with different gait patterns (provided in Appendix A). We then requested one member of the focus group to wear the sensors and test the feedback in real life (about 10 min) by simulating different gait patterns while the others made observations and engaged in an open discussion. The feedback was played back on a loudspeaker during this time. Following this, the focus group was requested to engage in a more structured discussion of the specific feedback algorithm; aspects, such as potential target groups, intuitiveness, ease of perception, and motivation potential were discussed. Five-minute breaks were taken after each FA. Finally, a 15-min wrap-up discussion was conducted to summarize the main outcomes of the session.

### 3.4. Data Analysis

The audio recordings were manually segmented into discrete statements, which were then transcribed, anonymized, and translated from Danish to English. The data were assessed for recurring ideas and/or experiences and an inductive thematic analysis was conducted using a ground-up approach. Based on the content, each recurrence was assigned to a theme, and a list of key takeaways was compiled. These were related to aspects such as feedback complexity, possible use cases, goal-orientedness, potential for gamification, and issues with the feedback algorithms.

### 3.5. Key Takeaways

*Use cases:* The overall assessment of the physiotherapists was that this type of swing phase feedback has the potential for use in addressing two major gait issues seen in the target group, i.e., asymmetry and low tempo (cadence). In terms of cadence, they explained that the general feedback philosophy of providing limb swing speed feedback through volume changes in the sound could potentially encourage patients to increase their walking tempo. This was deemed to align well with the goals of high-intensity gait training, and to foster self-training instead of relying on external therapist feedback. They judged that FA1 (wading) could potentially be used with patients having difficulties in swinging their shank forward, or those who exhibit insufficient hip flexion (with some modification to the algorithm).

*With (wading) sound feedback I think we can address tempo and asymmetry issues. Of course the sounds have to be tweaked to be more concrete.* [T3, Wrap-up Discussion]

*I wonder if it can be used for self training where the patient had headphones on, someone comes and clicks play and the tech gives feedback instead of us giving verbal feedback during the process.* [T2, Wrap-up Discussion]

*...there was a relationship between speed and volume. Here you could get the feedback to come up in tempo. It goes hand in hand with the ideal that we have to have higher intensity during training.* [T4, Wrap-up Discussion]

*Feedback complexity:* The physiotherapists estimated that having multiple layers of changes in the overall sound could be confusing, distracting, and cognitively challenging. For example, they felt that the melody (FA2, FA3) could interfere with the timbre and loudness. They added that background music over the wading sound (plus interactive music) or multiple instruments (flute plus violin) conveying thigh and shank SAVs might be too much information. In general, they asserted that simple feedback is preferable (at least to start with).

*Too much information so the patient is stuck in their head and forget their body, so cannot use it.* [T2, Synthesized Music]

*It is hard for us that don’t have brain damage to hear whether it is the octaves going around or the timbre that is changing.* [T4, Abstract Waveform]

*One also needs to be at a cognitive level to be able to distinguish between the feedback and the music that gets added later.* [T2, Wading]

*I think one could well add something complex onto the feedback but not immediately, maybe once the brain gets used to the sound.* [T5, Wrap-up Discussion]

*Goal-orientedness:* The physiotherapists expressed that having clear ‘sonic goals’ would make the feedback easier to use, such as explicit sounds to achieve or avoid. The visual feedback equivalent would display a green light to indicate when the desired step length has been reached. Hence, patients would have something to aim for (a reference), and feedback would be provided in relation to this reference. The goal would have had to be adjustable for different patients and it depended on the overall goal of the training. In this context, they stressed that using feedback sound types familiar to patients (e.g., known melody) would be desirable as they could provide a clear reference for what the patient should aim for. They added that instead of providing an explicitly unpleasant negative feedback signal (e.g., buzzer sound), it would have been more motivating to provide a clear auditory reward when a movement was performed ‘right’. It was unclear to them whether the adopted approach of providing feedback on the movement ‘as-is’ would suffice to achieve the intended purpose, or if the feedback would need to explicitly communicate task goals to the patients.

*If the sound clearly signals what you should and shouldn’t do, then there is a clearer goal in terms of what feedback is being given on.* [T3, Abstract Waveform]

*Talking about motivation, one should take care that one doesn’t provide a ’buzzer’ sound every time one makes a mistake.* [T4, Wrap-up Discussion]

*Something cool should happen when one walks normally. Right now not enough of a difference between right and wrong.* [T5, Synthesized Music]

*Customizability*. In general, the physiotherapists stressed the need for a customizable system that provides flexibility in several regards, drawing several comparisons to the C-mill system (https://www.motekmedical.com/solution/c-mill/, accessed on 28 February 2023) as different configurations might suit different types of patients. (1) It should be possible to adjust task goals to suit individual patients (e.g., knee flexion, hip flexion), including a system for ‘difficulty levels’; (2) for a given feedback type, there should be a set of sound types and sound effects to choose from, so as to reduce monotony over time.

*Would be nice to have different sound effects depending on what it is that we want to train.* [T2, Wrap-up Discussion]

*It would be good to have an option—difficulty level 1 2 3* [T1, Wading]

*Potential for gamification:* The physiotherapist who tested the algorithms in real-time saw some potential for gamification applications with such feedback due to the possibility of playing with the system to see what kind of sounds it could generate. This was the case for FA2 and (especially) FA1.

*Compared to the first one (abstract), I felt it was clearer in this example when I hit the correct swing with the shank. Because of that, I also felt more like playing with it and seeing what would happen when I took slow and high steps.* [T3, Wading]

*Issues with current feedback algorithms:* Several potential problems with the showcased algorithms were revealed during the session. They pointed out that the electronic-sounding nature of FA2 and the shrill sound of FA3 would make them difficult for patients to use over long training sessions. The physiotherapists expressed a preference for FA1 in this regard. However, the use of ambient music as the potential background for FA1 was not deemed appropriate as it was perceived as conducive to relaxation rather than moving one’s body. A criticism of FA1 was that even asymmetric gait sounded ‘pleasant’ and, thus, might not incentivize patients to change their gait pattern. This may have been because the sounds continued for a brief period after the completion of the swing phase, which was also the case for FA3.

*It is personal but I would have a hard time walking with this electronic sound for half an hour.* [T2, Abstract Waveform]

*That’s because the tone is held, when I walk fast the notes blend together.* [T3, Synthesized Music]

*The ambient music doesn’t go with the intention of walking and moving, I feel more like sitting on a beach and relaxing.* [T2, Wading]

## 4. Redesign Specifics

During the hands-on session, two serious problems that arose with regard to FA2 and FA3 were:1.The musical pitch changes controlled by the step modulo, which interfered with the sound properties representing the SAVs.2.The artificial and potentially annoying nature of their respective tonalities.

Upon subsequent experimentation, we found that simply disabling the pitch changes made the resultant feedback sounds highly monotonous and exacerbated their tonal issues. In the absence of a readily-applicable solution, we discarded FA2 and FA3 for the remainder of the study. Based on the inputs of the focus group, we refashioned FA1 into a general template that could be tweaked to provide direct wading feedback with the option of additional positive or negative reinforcement based on the set training goal(s). The plain wading algorithm was principally similar to FA1. We made improvements to the sound by using two medium-intensity flowing sounds for the thigh and shank, respectively, as opposed to a trickling sound for the thigh. Based on the focus group inputs, some new additions were made to facilitate the creation of goal-oriented training paradigms as follows:*Positive reinforcement (splash):* Here, the goal was to reward the patient with a clear impulsive splash sound each time their shank angular velocity crossed a configurable target value. The goal of the user would be to achieve the splashing sound during every step. The *difficulty level* could be adjusted by altering the target SAV value.*Negative reinforcement (urinating):* The principle was to provide the user with a water-based sound with a clear negative connotation if their thigh and shank angular velocities were too small in magnitude (configurable) during the swing phase. For this, we chose the sound of a person urinating in a toilet. This paradigm worked in such a way that the urinating sound was dominant when the swing velocity was low, and masked by the loud, broadband wading sounds when the velocity was high enough. The goal of the user would be to swing their lower limb segments fast enough to prevent the urinating sound from being audible during any step.

A schematic of the sensor processing is shown in Figure 6. As shown, two new optional sounds were added to the final mix (urinating and splash, both from the Freesound library); correspondingly, there were a total of four control signals $C_{1-4}$ and a set of new pre-processing operations:Envelope following: This operation was applied so as to make the falling edge of the amplitude envelope of the wading sounds more gradual and natural-sounding, while keeping the rising edge intact to ensure immediate responsiveness. We credit this approach to [62], and implemented the envelope follower as described in [58] (Chapter 12). The time constants used for the thigh and shank were different (370 ms and 670 ms, respectively).Rising edge impulsification: In order to randomly trigger one of three splashing sound samples when the $\omega_{S}$ thresholds were crossed, we converted the rising edges of the summed ON/OFF signals from the shanks to impulses of random height (see the $C_{4}$ branch in Figure 6).

The remainder of the data processing was similar to the first version, but one noteworthy point is that $C_{3}$ underwent a nonlinear transformation that was perceptually opposite to that of $C_{1}$ and $C_{2}$, such that the urinating sound was dominantly audible at low nonzero $\omega_{nT}$ and $\omega_{nS}$ values, whilst the flowing sound dominated at high values. All audio processing was done in REAPER. Video demos are provided in the online video repository.

## 5. Feasibility Study

After the redesign, we performed a feasibility study involving both patients and physiotherapists with the aim of exploring several aspects of their experiences when using the feedback, specifically patient enjoyment, self-perceptions of motor change, movement-feedback congruence, feedback naturalness, and therapist perceptions of motor change in the patient; we also collected any suggestions for improvement.

### 5.1. Participants

A total of nine hemiparetic patients (5M, 4F, aged 55.44 ± 16.08 years old) admitted to Neuroenhed Nord volunteered to participate in the study, along with their respective physiotherapists (seven in all, 13 ± 10.5 years of clinical experience). Informed consent was obtained prior to participation and all procedures were conducted in accordance with the Declaration of Helsinki. Relevant information about the patients is provided in Table 1.

### 5.2. Setup

The study was carried out in a ∼20 m long hospital corridor at Neuroenhed Nord, Regionhospital Nordjylland, Brønderslev, Denmark. The corridor was wide enough for the digital equipment and two experimenters (both authors) to be stationed along one side, whilst the remaining width was used by the physiotherapist and patient for gait training. The tech setup used was the same as for the hands-on testing, except that the feedback audio signal was sent to two separate playback devices:IKEA Eneby loudspeaker: This allowed everybody present at the training session to hear the feedback heard by the patient. The purpose of this was to help us adjust the system to the needs of the patient as well as allow the physiotherapist to monitor the feedback.Thomson radio frequency headphones (https://tinyurl.com/ynbmvvfc, accessed on 28 February 2023): These closed-back wireless headphones were provided to the patient in order to ensure that the feedback was provided with consistent sound quality, irrespective of the patient’s distance from the loudspeaker.

### 5.3. Procedure

The study was carried out individually with each patient–physiotherapist pair. The patient was escorted to the test location by the physiotherapist, and they were both briefed about the study, after which they provided their informed consent. The wireless sensors were then mounted on the thighs and shanks of the patient, and the therapist was requested to conduct a routine gait training session with the patient, the only novel elements being that the patient would have headphones on and that we would periodically apply different auditory feedback variants. The therapist was free to use the real-time feedback as they saw fit in terms of setting concrete training goals or providing instructions to the patient on the fly (e.g., “move more water with your right leg”, “create a splashing sound on both sides”, or “the sound from both sides need to be identical”). The patient had access to all of the usual assistive tools (manual support, rollators, support benches, walking sticks, etc.) during the session.

During the patient’s warm-up walk at the beginning, we recorded their baseline gait with no feedback applied, and used the data to manually set the angular velocity ranges for normalization in the software. In the case of substantial asymmetry, the ranges corresponding to the least affected thigh and shank were applied to both sides. We attempted to provide all three feedback variants (fixed order: neutral, positive, negative) to every patient, but this was not always possible, for instance when the patient became tired halfway. For each variant provided, we created recordings (2 min length on average) of the inertial data streams from all sensors at a 100 Hz sampling rate for subsequent analysis. Once the training was complete, the therapist and patient were invited to a 10-min interview where several aspects of the experience were discussed:Whether the patient enjoyed training with the feedback.Whether the patient felt that the feedback correlated well with their movements.Whether the patient noticed a change in their gait when using the feedback.Whether the patient found the wading sounds natural.Whether the therapist observed any change in the patient’s gait when using the different feedback variants.Whether the therapist had any suggestions for the improvement of the feedback.Any miscellaneous comments.

The interview was captured using a mobile audio recorder app on a Lenovo Yoga tablet for further analysis. All communication with the therapist and patient was done in Danish. The entire study took 45 min per patient.

### 5.4. Data Analysis

The interview recordings were analyzed in REAPER, where we first extracted discrete statements by the participants (therapists/patients), and then anonymized, labeled, transcribed, and translated them to English. The statements were then coded simplistically by a single analyst using an inductive (ground-up) coding approach, yielding a set of themes representing the key takeaways from the study. This was cross-checked by a second analyst, and the final list of themes was modified accordingly. Wherever the therapists expressed a change in the patient’s gait during one or more of the feedback conditions, we attempted to reconcile these statements with the motion data captured by the sensors during steady-state walking. Specifically, the angular velocity trajectories were plotted in MATLAB, and the raw inertial data were used to construct stick-figure videos of the patient’s gait using orientation information computed using the Madgwick gradient descent algorithm [63] integrated into our software. The next section presents a summary of the key findings from the study.

### 5.5. Results

#### 5.5.1. Patient and Therapist Experiences

*Patient-perceived advantages of feedback:* Several patients expressed perceived benefits of the feedback related to a sharpened awareness of their movement (three patients) and ability to hear their walking rhythm (two patients).

*I think it (splash sound) could help some people, and could be an OK extra action that tells people if you do this or that then you get this sound.* [P8]

*I think I would have a sharpened awareness if I trained with this....it could be good when I get used to synchronizing my sound to the movement, I could imagine it being a help.* [P1]

*It is the best thing I have tried so far, I felt completely safe and it was like I used my ears to hear the rhythm in how I walked.* [P5]

*I see the trick in having an rhythmic indicator or a scale for how consistent gait is, which gives good value.* [P7]

*Movement-feedback relationship and naturalness:* In terms of whether the sound and movement went together meaningfully, five patients explicitly stated that this was the case. Seven patients expressed that the feedback sounded natural to them. One patient stated that the feedback took some getting used to.

*The sound and movement went fine together.* [P1]

*In terms of sound-movement connection, they seemed synchronized and simultaneous.* [P7]

*We have all walked in water, and the sound that is heard when one moves their leg is something that I can relate to.* [P8]

*The reproduction is completely like listening to one trying to walk through water.* [P1]

*I actually think it faithfully sounded like it does when one walks through water.* [P3]

*In terms of naturalness, I think it was OK, I don’t think it can be more natural.* [P8]

*Moving and also understanding the sound at the same time is hard at the start, but it comes rather quickly.* [P6]

*Therapist-identified perceptual and cognitive issues:* Several therapists expressed that it was sometimes hard to hear when the movement was problematic (two therapists), particularly when the urinating sound feedback was used (two therapists).

*It could be even clearer when the patient goes completely against what they should be doing, like a really unpleasant sound, so they realize it’s all wrong.* [T1]

*I think it was a bit hard to differentiate when one does it right, but with the splash, I had an aha-moment when I could hear it.* [T6]

*I had a bit of difficulty hearing the pissing sound when you shifted over to it.* [T4]

*One could not hear the difference between the sounds, they were too close to each other, especially with the water and pissing sound (which sounds artificial).* [T8]

Two therapists were in favor of having a single focused sound rather than multiple (e.g., wading plus splash/wading plus urinating) due to the added cognitive burden.

*I think that it was the focused sound in the beginning that helped (plain) as opposed to having to achieve something—going fast (splash) or NOT doing something (urinating), because sometimes it is easier not having to think too much while walking.* [T3]

*It needs to be very usable, so having both the water sound and some other sound, which might be too many sounds.* [T9]

*Patient approaches to feedback use:* Upon reflection, three patients expressed the approaches that they adopted in terms of how they used (or would use) the sound feedback.

*I would go after the sound maybe in the long run, right now it was short enough that I would still have too much focus on walking correctly.* [P1]

*I was focused on trying to get them (left and right) to sound identical.* [P2]

*In terms of changing gait, it was the sound that I went after more than actually walking. It was much more intuitive in the sound in terms of whether I did it right.* [P7]

*Patient opinions on session-long feedback use:* In terms of whether patients would be able to tolerate the feedback for the typical duration of a training session (∼1 h) and whether the sound was irritating, opinions were mixed.

*It could get irritating at some point, I do not know, but I would easily be able to walk (with it) for an hour.* [P5]

*I won’t have a problem listening to this for a long time except the pissing sound.* [P7]

*Maybe it’ll be irritating to listen to for an hour. Max 30 min would be fine, but in an hour I would probably be stressed.* [P4]

*In the beginning one can be more motivated to get them to sound identical, but maybe later on, when one gets tired, it can be a bit unpleasant to hear that one cannot achieve the same result that one did when one started. I think that could irritate me a bit.* [P2]

*Therapist suggestions:* Two therapists provided suggestions on how the feedback could be improved, or offered new ideas altogether:

*The feedback could be adjusted even better to the patient so that they get that clear feedback, so that it can also be used with the better patients.* [T1]

*It could be nice to give feedback on how wide your steps are so as to challenge a patient to remain within a certain amount.* [T8]

*It could also be nice to try with cueing rather than feedback where you have to follow a rhythm rather than generating it yourself.* [T8]

*Patient Enjoyment:* When asked, the majority (six patients) expressed that it was fun to use feedback during their training.

*It was fun, something special, made sense for me, one could hear what each leg did.* [P2]

*It was fun to have the sound in my ears and hear myself walk.* [P4]

*My movement became more natural and we laughed “Can you hear me splashing?”* [P5]

*Negative experiences for one patient:* A single patient expressed general indifference and indisposition when describing their experience of using the sound feedback.

*I don’t know how it was to walk with this sound.* [P9]


*I didn’t feel that it was me that made the sound, it came from outside. [P9]*


*The sound was artificial.* [P9]

#### 5.5.2. Kinematic Characterizations

With several patients, the respective therapist observed changes in the patient’s gait during one or more feedback conditions, which the patients themselves may or may not have been conscious of. We found instances where these statements matched our findings from analyzing the SAV plots (Figure 7) and stick figure videos (available in Appendix A).

***Patient 2:*** The therapist and patient had the following to say:

*That was clearly the best I have seen you (patient) walk, both in terms of quality and tempo, don’t think I have seen anything comparable.* [T2]

*It was fun, something special...I don’t think I have walked so fast before.* [P2]

In the stick figure videos, we noted that relative to the baseline, the patient exhibited better control of knee deceleration during terminal knee extension in the *plain* and *splash* feedback conditions. In both cases, there was an increase in knee and thigh RoM, along with a higher cadence and more regular rhythm. The latter is also visible in the angular velocity plots (the top two panes, P2 (a) and (b) of Figure 7) with more steps taken within the same duration with greater consistency, as well as higher thigh and shank SAV peaks on the most-affected side, indicating a greater degree of visible symmetry.

***Patient 3:*** Here, the verbal comments were:

*You did change your movement in the first one (plain), I could see in terms of the left leg (most affected) as well as the rhythm you walked in. I was actually surprised by what the first sound did.* [T3]

*I don’t know whether I changed my movement.* [P3]

The stick figure video revealed that the patient exhibited a crouch gait with the most affected side. In the *plain* feedback condition, we noted an increase in knee RoM and a decrease in thigh RoM relative to the baseline. Knee deceleration was smoother throughout the swing phase and its termination. The SAV plots in Figure 7c show better gait regularity in the feedback condition in terms of both periodicity and shape consistency (particularly the most-affected shank), along with slightly better symmetry overall.

***Patient 4:*** The exchange during the interview was as follows:

*I felt like there were some improvements when you got the feedback.* [T4]

*I didn’t notice myself doing anything differently with my body.* [P4]

In the stick figure video, we noted that the baseline knee and thigh motion on the most-affected side was irregular and erratic. In the feedback condition, we noticed better temporal regularity and a slight increase in cadence, with more controlled and synchronous thigh and knee movement. Knee deceleration control during terminal swing was visibly better as well. In the P4 knee angular velocity plot in Figure 7d, the amplified negative peaks in the *plain* feedback condition indicate faster knee flexion, although no change in symmetry is visible.

There were also several cases where the data sources (verbal and kinematic data) were inconsistent with each other.

***Patient 9:*** In the case of this patient, there were some inconsistencies among the data sources.

*... it (feedback) got you to lift your leg more than you otherwise have.* [T9]

*I don’t think the sound got me to walk differently.* [P9]

In the stick figure video, we noted that the patient exhibited a high cadence and irregularities in thigh and knee coordination in the *baseline* as well as the *plain* condition, with no observable difference between them. Looking at the thigh SAV plot in Figure 7e (most closely connected to leg-lifting), we were unable to observe any corroboration of the therapist’s statement during the *plain* condition. In fact, when using the feedback, there appears to have been a deterioration in the symmetry of the thigh SAV.

There were three other cases where therapists reported noticing improvements in patient gait; the kinematic data failed to capture or corroborate this.

*I think that the patient adapted his walking to the sound so that he homed in on a correct walking pattern.* [T1]

*I think with the splash sound something happened, that we haven’t seen over the floor before, maybe only with treadmill.* [T6]

*I noticed that you got closer to being completely rhythmic, and also at one point your feet were closer to each other (less step width).* [T8]

## 6. General Discussion

In this study, we followed an iterative user-centered process to design a set of inertial sensor-based concurrent auditory feedback algorithms that informed patients on instantaneous variations in thigh and shank angular velocity during the swing phase. These were evaluated by a focus group of physiotherapists, on the basis of which we decided to proceed with only the reality-inspired wading algorithm for the purposes of this study. We then tested three variants of this algorithm in a feasibility study. As described in Section 2.3, the algorithms were designed based on rationale grounded in motor learning theories. As per [42], our approach allowed us to rapidly design and evaluate multiple feedback designs in terms of usability and acceptance, helping taper a relatively broad set (abstract, ecological, musical, and their sub-variants) to one promising candidate. Although heterogeneous, the overall results showed the plain wading FA to have considerable potential in terms of meaningfulness, user acceptance, and potential for eliciting kinematic change. On the other hand, several issues arose in connection with the goal-oriented paradigms (especially the urinating sound). Our user-centered methodology, aside from being recommended by past research [36,38], also provided insight into good feedback design practices in the context of hemiparetic patients. We now reconcile our overall findings with the feedback requirements defined in Section 2.4.

### 6.1. Angular Velocity Measurement, Clinical Compatibility, and Feedback Actuation

By directly sonifying smoothed SAVs, we were able to provide concurrent feedback without any perceivable latency, which is important for biofeedback systems [47] (Chapter 1), and for promoting multisensory integration [19]. Based on past experiments with a similar system [64], we estimated our overall loop delay to be <100 ms, which is well below the average auditory reaction time of 160 ms [47] (Chapter 5). Motion sensing was entirely conducted by using inexpensive wearable IMU sensors that have been shown to be reliable in gait tracking [43,45,46], implying that similar feedback algorithms may be realizable even in existing mobile devices [65]. The patient-specific configuration of swing phase angular velocity ranges and subsequent normalization applied in all feedback algorithms demonstrate how such swing phase capture and feedback can be made adaptable to suit the inherent variability in the target group [1]. We also demonstrated how the feedback could potentially be integrated into the context of a real-life gait training session, although several questions arise at this stage. In our feasibility study, the system setup, device mounting, calibration, and feedback configuration were carried out by researchers. In practice, all of these tasks will have to be performed by the therapist, and we judge our system in its current state to be too complex for efficient real-world use. A streamlined software interface (possibly on a mobile device) with a simplified sensor connection procedure will be necessary in order to realize a usable biofeedback toolbox for clinical use [8].

The choice of a suitable feedback actuation device for use in a clinical overground training setting is still an interesting question [10]. Real-life training locations can vary widely in terms of shape, size, reverberant properties, and ambient noise levels; all of these factors can potentially interfere with patients being able to clearly perceive the feedback stimuli [66]. It is also important for the therapist to be able to hear the feedback generated by the patient in order for them to be able to reconfigure or troubleshoot it. Finally, the therapist needs to be able to communicate verbally with the patient without difficulty. Using a sufficiently powerful loudspeaker as the sole actuator can ensure the collective audibility of the feedback while also allowing conversation between the patient and the therapist. However, the fixed location of the loudspeaker will mean a reduction in both loudness and the ratio of direct-to-reverberant sound heard by the patient as they move further away from the speaker. This would not only make the contained information harder to perceive, understand, and act on, but also potentially increase the cognitive load [67]. Our solution of providing the feedback through wireless radio frequency headphones was advantageous in the following ways: (1) the headphones generated negligible propagation delays (would have been substantially higher for Bluetooth), (2) the feedback sound heard by the patient maintained its characteristics irrespective of how far the patient walked, and (3) the proximity of the actuator minimized the effects of the corridor acoustics on the stimulus heard by the patient. By splitting the audio output feed between the headphones and loudspeaker, the therapist was able to hear the feedback and judge how it reflected walking characteristics. However, the use of closed-back headphones interfered more than once with therapists attempting to communicate verbally with patients. Given all of this, the ‘best’ actuator combination remains unclear.

### 6.2. Feedback Design

The hands-on physiotherapist testing and feasibility study yielded several useful insights that can guide auditory feedback design processes targeting hemiparetic patients in particular as well as neurological patients at large. The general design philosophy of representing angular velocity as energetic variations in the sound was deemed by the focus group to align well with clinical goals, and as expected, ended up being fairly easy for the patients to match with the perception of their movements, implicitly enhancing their awareness of symmetry and tempo. This matches findings from our past work on the sit-to-stand transition [68], supporting our conclusion that this general velocity-energy mapping is an intuitive one. Having said that, there is a clear limit to the level of stimulus complexity and layering that hemiparetic patients can be expected to tolerate. Based on existing literature [59,60], we attempted to incorporate musical structures into FA2 and FA3 in an attempt to enhance motivation during training. However, the focus group members found that these additional layers (background music, pitch changes) of stimulus evolution were confusing as they perceptually interfered with the sound variations corresponding to the feedback. This aligns perfectly with existing research on perceptual interactions between sound properties, especially pitch and loudness [69]. However, stimulus novelty and unpredictability are important in sustaining user feelings of motivation and reward in sonic interaction [59]. Our attempt to address this by imbuing FA1 with an ambient music element was not received well by the focus group. The balance between predictable feedback behavior and musical evolution over time remains elusive in this context. During the feasibility study, there were instances of patient and therapist confusion brought about when several feedback sounds were layered (e.g., urinating or splashing sounds layered with the plain wading sound), which may have been exacerbated by the overlapping spectral content of the sounds. The general recommendation of the focus group was to start with a very simple feedback sound and, if relevant, gradually add layers once the patient became familiar with it. This raises the question of how best to apply goal-oriented paradigms (which was suggested by the therapists) without overloading the feedback stimulus with information. One option could be to drop the plain wading sound in these cases so that the patient can focus exclusively on the positive or negative feedback stimuli, similar to designs from past studies [25,70].

In terms of sonic aesthetics, the artificial-sounding timbres of FA2 and FA3 were not deemed by the focus group to be usable for the duration of a typical training session. Instead, they clearly favored the relatable sound texture of FA1 (wading) and estimated that it would be well-tolerated by patients. Their estimation was validated during the feasibility study, where several patients stated that they would not mind listening to this feedback for the typical duration of training. We attributed this significant finding to several factors, with perhaps the most important being that the sound feedback was based on a common real-life experience typically associated with nature. Such sounds have been shown to be calming in clinical ecosystems [71]. Moreover, the majority of the patients found the wading feedback very natural-sounding. This may have led to the patients being (1) receptive towards the wading sound itself, and (2) able to implicitly and intuitively link their own movement characteristics to those of the sound. This is supported by our own past finding that the perceived naturalness of auditory feedback is inversely correlated with perceived intrusiveness [72], as well as studies that have found that ecological stimuli are easier to interpret and act on than abstract ones [33,34], aside from being more motivating [6]. Broadly speaking, our findings support the use of ecological metaphors suggested by an accumulating body of literature in auditory biofeedback and sonification design at large [17,32,35]. Our ability to design a natural-sounding wading simulation is primarily attributed to having had access to a representative hemiparetic gait dataset and sonification software with real-time control over important components of the data-sound mapping function (smoothing, scaling, transfer function shape, etc.) [73].

### 6.3. Kinematic Changes

Due to the exploratory and non-randomized-controlled protocol followed in the feasibility study, we are unable to conclude any causal relationship between the feedback and any gait changes relative to the baseline condition (such as those shown in Figure 7). However, we believe that a rigorous effect study is warranted on the basis of the dramatic differences seen for some of the patients in terms of cadence (P2), SAV symmetry (P2, P3), gait rhythmicity (P3), and joint rotational velocity (P4) between the baseline and feedback conditions. The differences observed by some therapists largely correlated with kinematic analyses of the angular velocity plots and the stick figure videos, even if the patients themselves were unconscious of any change in their movement quality. Based on the interviews, it is plausible that the feedback acted upon these patients in the following ways:The differences in sound generated by the least- and most-affected sides enhanced their awareness of their walking asymmetry, which prompted them to modify their movements to make the two sides sound more identical.The synchronous and repetitive rise and fall of the wading sound envelope heightened the patients’ awareness of any rhythmic irregularities in their gait, prompting them to modify their gait rhythm to make it more periodic.

In the remaining patients, especially those with higher FAC scores and better overall mobility, the prevalence of any differences in gait quality between conditions was much harder to detect, both visually and graphically. Based on the interview data, this could be because (1) severe asymmetry was easier for patients to perceptually detect in the feedback than mild asymmetry, (2) faster-walking patients generated the feedback sound at a higher tempo, making it harder for them to discern and react to rapid changes in the sound characteristics, and (3) the slow time-constants of the envelope follower smoothed out some of the rapid angular velocity modulations suggestive of the pathological gait (see Figure 2), rendering them inaudible. As sharpening the temporal behavior of the wading sound would probably compromise its perceived naturalness [72], it may well be that this feedback algorithm is best suited to slow-walking patients, whilst a different algorithm must be designed for better and faster walkers exhibiting mild asymmetry. We also noted that P9 exhibited worse thigh symmetry in the feedback condition, but this may have been the result of the perceptual and cognitive challenges that also led to this patient generally not being receptive to feedback. This is a known caveat of using sound- and music-based rehabilitation technology [38].

Nevertheless, the presented results show that for a subset of patients, this form of feedback design may bring immediate positive changes in walking kinematics. As the feedback is tightly and ecologically tied to limb movement, one may hypothesize that it can permanently be integrated into the patient’s motor model based on existing theories of multisensory integration and learning [17,18,19,20,49] as well as past studies employing comparable feedback designs in other motor tasks [74]. As the plain wading FA does not contain any form of error information, the potential for an undesired guidance effect is minor [6,52], although this will have to be tested for any goal-oriented paradigms.

### 6.4. Limitations and Future Work

As this was an exploratory study that focused on design and feasibility, we did not conduct a randomized-controlled trial that included a no-feedback age- and gender-matched control group to test the effects of the feedback on gait quality. This is part of our planned future work. Therefore, we cannot determine whether the observed gait changes were due to the feedback or simply the patients becoming more accustomed to the task during the training session. Our future work will also involve investigating the ideal stimulation frequency during training to optimize the feedback’s effects on gait patterns. Our kinematic analysis was limited to characterizations based on angular velocity trajectories and reconstructed body segment traces. In both evaluations, the interview activities were not blinded, which may have possibly induced social desirability biases in the participants. Future work will involve implementing and testing a swing phase FA that can clearly convey swing phase issues in fast-walking, high FAC scoring patients (e.g., using vocal patterns [75]), as well as developing a new FA adapted for hip flexion training as suggested by the physiotherapists. We will also have to rework or discard the goal-oriented feedback paradigms in the wading sonification algorithm, as the current versions have several issues. An interesting addition would be the integration of an automated adaptive configuration of angular velocity ranges based on short-term data from the least affected side. Based on the findings of [16,24], it may be beneficial to provide patients with the sound of a "normal" wading gait and instruct them to imitate the corresponding movement. It may also be worth exploring concurrent feedback in the frontal plane (e.g., related to step width) based on one of the therapist’s suggestions. Finally, a comparison of our feedback with similar designs in the visual or haptic domain would help us ascertain the most effective modality for concurrent swing phase feedback. Along these lines, it would be interesting to experiment with combinations of modalities to investigate whether multisensory integrative mechanisms can be invoked to a greater degree to enhance motor learning [6,20]. The wading algorithm could, in principle, be supplemented by congruent visual feedback (perhaps in a virtual reality environment) as well as synchronized vibrotactile feedback.

## 7. Conclusions

In our study, we established the feasibility of providing hemiparetic patients with concurrent swing phase auditory feedback on thigh and shank kinematics using ecological sounds that evoke real-life wading experiences. Based on our results, it is evident that inexpensive wearable gyroscopes have significant untapped potential to serve as a reliable data source for biofeedback applications with only basic pre-processing. Further research is necessary to establish the effectiveness of the feedback and define specific use cases for each of the variants, which may require some redesign. Overall, we believe that the clinical potential of this form of concurrent swing phase feedback warrants further scientific investigation.

## Figures and Tables

**Figure 1 sensors-23-03964-f001:**
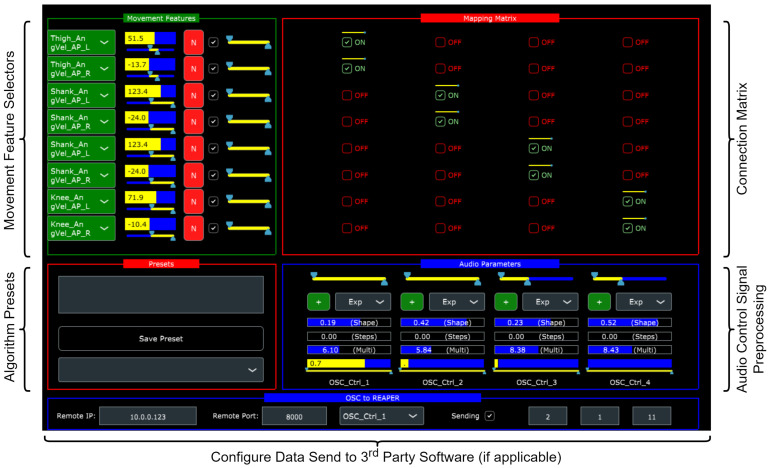
The parameter mapping interface of the software platform with the main functions labeled.

**Figure 2 sensors-23-03964-f002:**
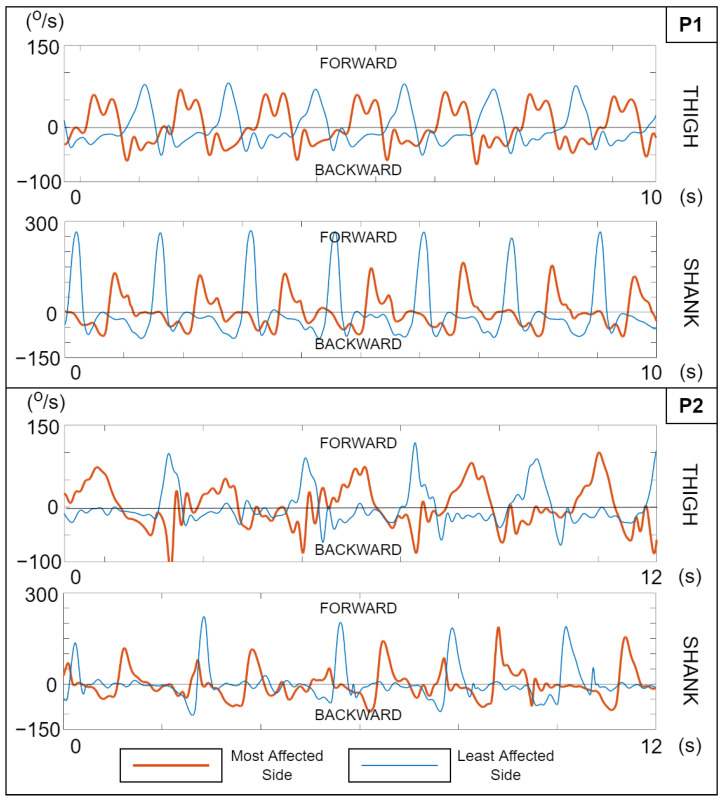
Example SAV plots from two patients with different levels of motor impairment. The thick red traces depict the most affected side in each case. The horizontal axis indicates the time in seconds, and the vertical axis shows the angular velocity in deg/s. The positive vertical direction indicates the segment rotation in the forward direction and vice versa.

**Figure 6 sensors-23-03964-f006:**
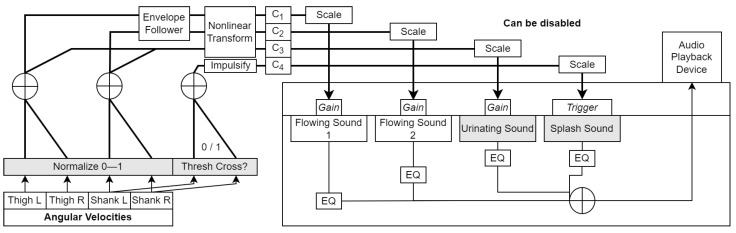
Block diagram of the modified wading algorithm. Grey boxes indicate modifiable parameters. *EQ* refers to the spectral equalization applied to each sound for tone adjustment. $C_{1}$–$C_{4}$ are the audio parameter control signals generated from the movement features.

**Figure 7 sensors-23-03964-f007:**
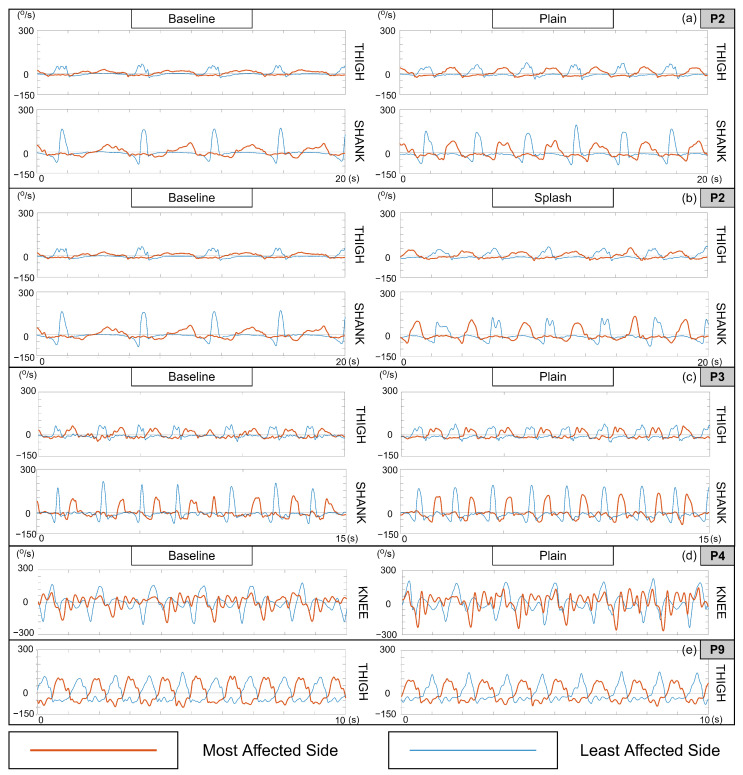
Kinematic plots of steady-state walking excerpts (**a**–**e**) during baseline and feedback conditions for selected patients (P2, P3, P4, and P9). In the thigh and shank SAV plots, the positive Y direction indicates forward rotation. In the knee plots, it indicates extension.

**Table 1 sensors-23-03964-t001:** Summary of the nine patients enrolled in the feasibility study. FAC = functional ambulation category score, FIM-Cog = functional independence measure (cognitive domain), n/a = not available.

ID	Gender	Age	FAC	FIM-Cog	Location of Stroke/Injury	Walking Aids
P1	M	62	4	29	Left middle cerebral artery	None
P2	F	43	2	25	Cerebellum	High rollator walker
P3	F	78	1	26	Right middle cerebral artery	Therapist support, Walking stick
P4	M	25	5	n/a	Left thalamus	None
P5	F	68	1	20	Right thalamus	Support bench
P6	M	52	5	23	Middle cerebral artery	None
P7	M	53	5	31	Right medulla oblongata	None
P8	M	51	4	31	Right subdural, Bilateral subarachnoid	None
P9	F	67	1	21	Right basal ganglia	None

## Data Availability

The data collected and analyzed during the study are available from the authors on request.

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
