# Peer review of "Designing Ecological Auditory Feedback on Lower Limb Kinematics for Hemiparetic Gait Training"

_sensors, 2023, doi:10.3390/s23083964_

Round 1
Reviewer 1 Report
Authors present a very interesting study about a user-centered design approach. Kinematic data were recorded from 15 hemiparetic patients to study gait training.
It is well-documented the role of Rhythmic auditory stimulation on gait training in patients with Parkinson's disease: the selection of an adequate frequency of stimulation can optimize their effects on gait pattern [10.1016/j.clinph.2019.07.013].
This “ecological” version is interesting but presents several limitation. I greatly appreciated the presence of the limitation session but I recommend authors to test this preliminary observation with a control group.
Moreover, I recommend to prove the effectiveness of the inexpensive sensors by a validation with an expensive gold standard like gait analysis (10.3390/ijerph192013440).
Author Response
Thank you for your comments on the paper, “Designing Ecological Auditory Feedback on Lower Limb Kinematics for Hemiparetic Gait Training” (ID: sensors-2283729). We have addressed all the comments as detailed below. We greatly appreciate the time and effort you have put into the peer review process, and believe that the paper has been improved as a result.
We have organized the comments as a numbered list, quoting them as-is. Underneath each one, we provide our response and/or description of how the comment was addressed. We then specify the exact location (page and line number) where the respective manuscript changes were made, if any. Please refer to the supplemental manuscript provided herewith for line numbers, where these changes are also highlighted.
- Authors present a very interesting study about a user-centered design approach. Kinematic data were recorded from 15 hemiparetic patients to study gait training.
Response: Thank you for the positive comment.
- It is well-documented the role of Rhythmic auditory stimulation on gait training in patients with Parkinson's disease: the selection of an adequate frequency of stimulation can optimize their effects on gait pattern [10.1016/j.clinph.2019.07.013].
Response: We acknowledge that stimulation frequency is an important variable in the administration of rhythmic auditory stimulation. Although this is different from our application (real-time feedback), it is true that it is important to find the optimal feedback frequency for a given scenario. We have added a line to this effect to the future work.
Location of Change in Manuscript: p. 24, lines 955-956
- This “ecological” version is interesting but presents several limitations. I greatly appreciated the presence of the limitation session but I recommend authors to test this preliminary observation with a control group.
Response: Thank you for the suggestion. We have added a line acknowledging this to the future work.
Location of Change in Manuscript: p.24, lines 951-953
- Moreover, I recommend to prove the effectiveness of the inexpensive sensors by a validation with an expensive gold standard like gait analysis (10.3390/ijerph192013440).
Response: Sensor reliability and validity are certainly important in biofeedback applications. There have been multiple past studies that compared our inertial measurement unit (MPU9250) with optical systems in terms of gait measurement reliability in healthy and patient populations. These studies concluded that the MPU9250 provides sufficiently reliable and accurate measurements. We have mentioned this and cited four studies (new references 43-46) in the methods and discussion sections.
Location of Change in Manuscript: p. 3, lines 122-125, p. 21, lines 820-821, p. 26, lines 1095-1102
Reviewer 2 Report
This paper has designed Ecological Auditory Feedback (EAF) for the gait actions of subjects under wearable exoskeletons. This paper is well-presented and easy to follow.
Some minor issues:
1) How is the auditory feedback close to the cognitive purposes of subjects? Is there any mapping based on data-driven approaches?
2) Other factors: some other stimulation feedback may be considered to have potential influence, are they more efficient than auditory feedback?
3) How about multiple modalities including auditory feedback?
Author Response
Thank you for your comments on the paper, “Designing Ecological Auditory Feedback on Lower Limb Kinematics for Hemiparetic Gait Training” (ID: sensors-2283729). We have addressed all the comments as detailed below. We greatly appreciate the time and effort you have put into the peer review process, and believe that the paper has been improved as a result.
We have organized the comments as a numbered list, quoting them as-is. Underneath each one, we provide our response and/or description of how the comment was addressed. We then specify the exact location (page and line number) where the respective manuscript changes were made, if any. Please refer to the supplemental manuscript provided herewith for line numbers, where these changes are also highlighted.
- This paper has designed Ecological Auditory Feedback (EAF) for the gait actions of subjects under wearable exoskeletons. This paper is well-presented and easy to follow.
Response: Thank you for this positive comment.
- Some minor issues:
- How is the auditory feedback close to the cognitive purposes of subjects? Is there any mapping based on data-driven approaches?
Response: Our mappings are not directly data-driven per se, but they are based on known embodied associations between movement and sound (velocity ∝ sound intensity) and make use of relatable sound structures from real-life experience (e.g. wading). We evaluated the perceptual and interactive qualities of the wading mapping in a recent study (under review), which showed that it was sufficiently realistic-sounding and natural to interact with. Since this is already mentioned in the text, we have not made any changes at present.
- Other factors: some other stimulation feedback may be considered to have potential influence, are they more efficient than auditory feedback?
Response: The choice of modality for a feedback application depends on the motor feature to be fed back as well as the consideration of which modality enables the most precise perception of that feature. If gait cadence (tempo) and periodicity are the focus, then the auditory modality is probably most suitable. However, if spatial symmetry is the focus, a visual display may provide more accurate information, albeit at the cost of diverting visual attention from the walking task. In order to compare overall efficiency in conveying a kinematic variable, the modalities would have to be directly compared. This could be part of future work, and a line to this effect has been added to the discussion.
Location of Change in Manuscript: p. 24, lines 970-972
- How about multiple modalities including auditory feedback?
Response: There is a great deal of potential in combining multiple modalities to enhance multisensory integration and, in turn, motor learning. In our context, it could certainly be worthwhile to integrate the visual and haptic modalities into the wading feedback algorithm to create a more compelling multimodal interaction. We have made a brief addition to this effect to the discussion.
Location of Change in Manuscript: p. 24, lines 972-976
Round 2
Reviewer 1 Report
Thank You